# Priorities of patients with multimorbidity and of clinicians regarding treatment and health outcomes: a systematic mixed studies review

Harini Sathanapally ![ORCID],[1] Manbinder Sidhu,[2] Radia Fahami,[1] Clare Gillies,[1] Umesh Kadam,[1] Melanie J Davies,[1] Kamlesh Khunti ![ORCID],[1] Samuel Seidu[1]

[1]University of Leicester Diabetes Research Centre, Leicester, UK
[2]School of Social Policy, Health Services Management Centre, University of Birmingham, Birmingham, UK

**Correspondence to**
Dr Harini Sathanapally;
hs333@student.le.ac.uk

## ABSTRACT

**Objectives** To identify studies that have investigated the health outcome and treatment priorities of patients with multimorbidity, clinicians or both, in order to assess whether the priorities of the two groups are in alignment, or whether a disparity exists between the priorities of patients with multimorbidity and clinicians.

**Design** Systematic review.

**Data sources** MEDLINE, EMBASE, CINHAL and Cochrane databases from inception to May 2019 using a predefined search strategy, as well as reference lists containing any relevant articles, as per Preferred Reporting Items for Systematic Reviews and Meta-Analyses and Cochrane guidelines.

**Eligibility criteria** We included studies reporting health outcome and treatment priorities of adult patients with multimorbidity, defined as suffering from two or more chronic conditions, or of clinicians in the context of multimorbidity or both. There was no restriction by study design, and studies using quantitative and/or qualitative methodologies were included.

**Data synthesis** We used a narrative synthesis approach to synthesise the quantitative findings, and a meta-ethnography approach to synthesise the qualitative findings.

**Results** Our search identified 24 studies for inclusion, which comprised 12 quantitative studies, 10 qualitative studies and 2 mixed-methods studies. Twelve studies reported the priorities of both patients and clinicians, 10 studies reported the priorities of patients and 2 studies reported the priorities of clinicians alone. Our findings have shown a mostly low level of agreement between the priorities of patients with multimorbidity and clinicians. We found that prioritisation by patients was mainly driven by their illness experiences, while clinicians focused on longer-term risks. Preserving functional ability emerged as a key priority for patients from across our quantitative and qualitative analyses.

**Conclusion** Recognising that there may be a disparity in prioritisation and understanding the reasons for why this might occur, can facilitate clinicians in accurately eliciting the priorities that are most important to their patients and delivering patient-centred care.

## Strengths and limitations of this study

► This is the first systematic review to assimilate and compare the findings of existing literature on the health outcome and treatment priorities of both clinicians treating and patients living with multimorbidity.

► We have included papers using both qualitative and quantitative methodologies and have been able to explore patterns and relationships in the findings, thus creating a comprehensive and well-rounded systematic review.

► Our findings facilitate clinicians in understanding both how and why the health outcome and treatment priorities of their patients with multimorbidity might differ from their own priorities.

► Meta-analysis of the quantitative studies was unfeasible as there was a large variation in the tools used to ascertain priorities, and we have attempted to mitigate this by using a well-described and transparent method of narrative synthesis.

► A number of our included quantitative studies did not use prevalidated tools to ascertain priorities, leading to a risk of measurement bias.

**PROSPERO registration number** CRD42018076076.

## INTRODUCTION

Multimorbidity, defined as the coexistence of two or more long-term conditions,[1] is a global problem,[2] which has become the norm across high-income countries[2–5] and becoming increasingly prevalent in middle-income and low-income countries.[2 6 7] Guidelines for the management of chronic diseases are often single disease orientated, and can lead to confusion and complications when applied to patients with multimorbidity.[8] Patients with multimorbidity have an increased risk of adverse drug-related events as a result of high

levels of polypharmacy and receiving uncoordinated care from multiple healthcare providers.[9] These patients have a poorer health-related quality of life,[10] poorer functional status[11] and greater psychological distress.[12] As a result, understanding and finding better strategies to facilitate the management of patients with multimorbidity has been identified as a priority for health research.[13]

Key to the effective management of multimorbidity is using patient-centred care and shared decision making to set management goals that are acceptable to both the patient and the clinician.[14] Incorporating the priorities of patients in relation to treatments and health outcomes is integral to this process.[15–17] However, previous research has shown that while doctors recognise the importance of eliciting and incorporating the priorities of their patients with multimorbidity, they do not always engage with this process in real-world settings, and find eliciting patients' priorities to be difficult.[18 19] Previous research, completed in a single disease context, has shown that the treatment and health outcome priorities of patients and clinicians can differ,[20–22] and some studies have highlighted a gap between what doctors' perceive to be the priorities of their patients, and the actual priorities of their patients.[23–25]

This systematic review aims to identify studies that have investigated the health outcome and treatment priorities of patients with multimorbidity, clinicians or both, in order to assess whether the priorities of the two groups are in alignment, or whether there is a disparity between the priorities of patients with multimorbidity and clinicians.

## METHODS
### Search strategy
A comprehensive search strategy (online supplementary appendix 1),was developed using guidance for the best practice[26] and input from academic librarians at the University of Leicester. The search strategy was used to search MEDLINE, EMBASE, CINHAL and COCHRANE databases from inception to May 2019, as well as searching reference lists for any relevant articles based on Preferred Reporting Items for Systematic Reviews and Meta-Analyses and Cochrane guidelines.[26–28] We undertook a scoping search using google scholar using our key terms (Patient*; Priorit*; Clinician, Physician, Doctor, General-practitioner, Family-practitioner; Multimorbidit*; Multi morbid*) to identify relevant grey literature. Citations were stored using Refworks. We have presented our process of article selection in figure 1.

We included studies reporting the health outcome and treatment priorities of adult patients with multimorbidity[1] and/or clinicians, in relation to patients with multimorbidity. Studies which did not specify the definition of multimorbidity as 'two or more chronic conditions'[1] in their inclusion criteria, but had a sample patients representative of being diagnosed with multimorbidity (ie, with a minimum of two chronic conditions) were also included. There was no restriction by study design, and we included studies using quantitative and/or qualitative

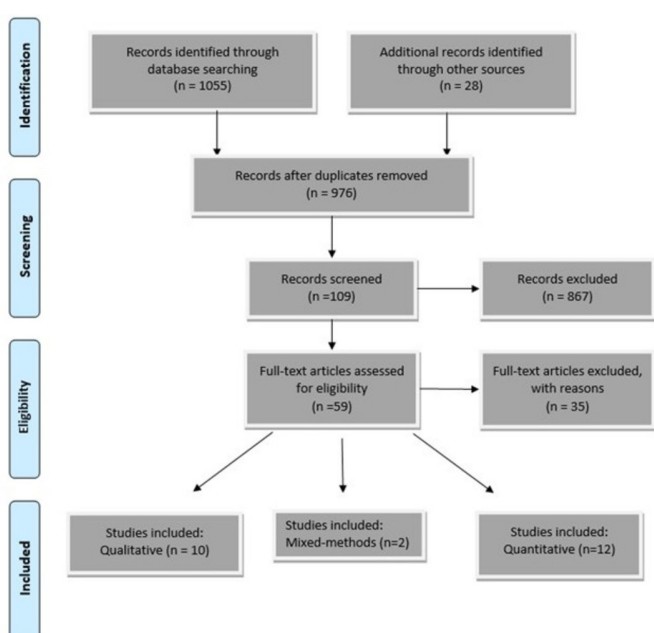

**Figure 1** Flow diagram to illustrate process from literature searching to selection of studies for inclusion.[28]

methodologies. We excluded studies not published in English language, studies with participants aged under 18 years and studies focusing on a single disease area.

### Patient and public involvement
Patient and public involvement was not applicable in the design, conduct or reporting of this review.

### Study selection
The titles and abstracts of all articles identified by the literature search were assessed independently and in duplicate by two reviewers (HS and RF). Studies that did not meet inclusion criteria were discarded. Full text of selected articles was retrieved and assessed to determine if they met the inclusion criteria, and those studies which met the inclusion criteria were included in the review. Any discrepancies regarding eligibility of an article were discussed, and consensus reached with MS and SS.

### Methodological quality assessment and data extraction
Data were extracted using standardised data extraction forms by a single reviewer (HS), and these were checked independently for accuracy by a second reviewer (SS). The reported health outcome and treatment priorities of study participants were the key outcomes that were extracted.

Quality assessment was carried out in parallel with the data extraction process. For the quantitative studies, due to the heterogeneity of study design, we used the Appraisal tool for Cross-sectional Studies (AXIS tool) for assessment for the cross-sectional studies,[29] the Newcastle-Ottawa scale for assessment of the longitudinal observational and cohort studies,[30] and the Cochrane collaboration's risk of bias tool for assessment of randomised controlled trials.[31] For the qualitative studies, we used the Critical Appraisal

Skills Programme (CASP) checklist for appraisal of qualitative research.[32] For the two mixed-methods studies, we used the AXIS tool[29] to assess the quantitative aspects of the study (both cross-sectional in study design), and the CASP checklist for qualitative research,[32] to assess the qualitative aspects of these studies.

## Data synthesis

We decided a priori not to carry out a meta-analysis due to the heterogeneity of the quantitative studies. Therefore, we have taken a narrative synthesis approach, described by Popay *et al*[33] to synthesise our quantitative findings. Our approach consists of three key steps:

1. Development of a preliminary synthesis in which study characteristics and descriptions are collated and findings presented in a summary table.
2. Exploring relationships in the data between study characteristics and their findings, as well as between the reported findings across different studies with explanations considered where relationships were identified.
3. Assessing the robustness of the synthesis using quality assessment tools to guide conclusions and identify directions for clinical practice.

Qualitative studies were synthesised using a meta-ethnography approach,[34 35] which consisted of careful reading of the papers, extracting information regarding the context of the study and findings. Key concepts arising from each paper were also identified, with preservation of the terminology used by the authors where possible to ensure accurate representation of the findings of the original studies. The key concepts across the papers were then translated using a table summarising the studies, their findings in relation to the key concepts and the second order interpretations of the authors, which enabled the exploration of any relationships and differences. The translations were then synthesised using a table containing the first order and second order interpretations for the key concepts across the studies, which then led to the development of further, third order interpretations by reviewers.[34 35]

From the results of our narrative synthesis of the quantitative studies and meta-ethnography of the qualitative studies, we considered how the findings of the two syntheses complement one another, particularly where our qualitative findings may provide possible explanations for our quantitative findings. The outcome of this process is described in the discussion section.

## RESULTS

### Overall study characteristics

Our search resulted in the identification of 24 studies for inclusion, which comprised 12 quantitative studies, 10 qualitative studies and two mixed-methods studies. The characteristics of all of the included studies are described in table 1. The included studies had all been conducted in high-income developed countries, including Canada,[36 37] USA,[38–44] Netherlands,[45 46] Australia,[47 48]

UK,[49–51] Germany[52–55] and Switzerland.[56–58] Sample sizes ranged from 15 to 1169 patients and 5 to 92 clinicians in the quantitative studies, and 15 to 146 patients and 4 to 19 clinicians in the qualitative studies.

### Summary of quality assessment

The outcome of quality assessment based on each of the aforementioned tools is summarised in online supplementary appendix 2. The majority of the quantitative studies were cross-sectional in design,[36 39 40 45–47 53 54 56–58] including the quantitative elements of the two mixed-methods studies. The other studies included one cohort study[44] and one randomised controlled trial.[52] The cross-sectional studies were of moderate quality, with a number of studies having small sample sizes.[40 45 46 54] The sample sizes of clinicians in most of the cross-sectional studies were particularly small, ranging from of 9 to 157 clinicians,[45 46 54 57] which impacts on the generalisability and application of their findings. We noted that a number of the studies did not use prevalidated questions and tools to ascertain priorities,[36 54 56–58] leading to a degree of subjectivity in the way in which priorities were ascertained, and the risk of measurement bias which again impacts on the generalisability of their findings.

The majority of the qualitative studies, including the qualitative aspects of the two mixed-methods studies, used interviews for data collection (n=8). Two studies used focus groups,[41 55] one study used a combination of focus groups and interviews[49] and one study used the nominal group technique.[48] The qualitative studies were of good quality, with appropriate use of qualitative methodology and transparent descriptions of the data analysis processes. Three studies only gave a limited description of their analytic process,[47–49] with two of these studies[47 48] and one mixed-methods study,[45] not presenting any quotes.

## QUANTITATIVE SYNTHESIS

Within our quantitative synthesis, we found that the studies focused either on the overall state of the patients' health, the problems posed by different chronic disease groups or the patients' treatment regimens. Some of the quantitative studies elicited patient and/or clinician priorities as part of an intervention.[46 52] Therefore, in order to reduce the risk of bias from the interventions, we included only the preintervention results from these studies.

### Health outcome priorities

Four studies reported patient priorities of overall health outcomes using a 'health outcome prioritisation tool',[39 40 45] which is a visual analogue scale requiring the following health outcomes to be given a score out of 100: 'maintaining independence'; 'staying alive'; 'pain relief'; 'symptom relief'. Maintaining independence was the outcome that had the highest importance after a pooling of the most important rankings from the four studies, followed by 'staying alive' (table 2). For clinicians'

**Table 1** Characteristics of all of the included studies in order of reference

**Quantitative**

**Health outcome priorities**

| Author and year of publication | Setting | Study type | Study aims | Target group and no of participants (n) | Outcomes measured |
|---|---|---|---|---|---|
| Quantitative | | | | | |
| Health outcome priorities | | | | | |
| Fried 2011[39] | USA—three senior centres and one assisted living facility | Quantitative: Cross-sectional Study. | To explore the use of a simple tool to elicit older persons' health outcome priorities. | All volunteers included (n=357). | The prioritisation by participants of 4 universal health outcomes, namely:<br>▲ keeping alive<br>▲ maintaining independence<br>▲ reducing or eliminating pain<br>▲ reducing or eliminating other symptoms. |
| Fried et al, 2011[40] | USA—recruited from participants in a larger study, where they had been recruited from age-aggregated community housing[74] | Quantitative: Cross-sectional Survey. | To determine the feasibility of using a simple tool to elicit the preferences of older persons based on their prioritisation of universal outcomes. | Patients aged 65 and over with a known diagnosis of hypertension or use of antihypertensive medications, and having a known risk of falls (n=81). | >Rankings given by participants to four universal health outcomes in the outcome prioritisation tool:<br>▲ -keeping alive<br>▲ maintaining independence<br>▲ reducing or eliminating pain<br>▲ reducing or eliminating other symptoms<br>>Feasibility of the use of outcome prioritisation tool. |
| Mantelli et al, 2018[57] | Switzerland—GPs working in Switzerland who had previously taken part in case-vignette studies | Quantitative: Cross-sectional Survey. | To determine whether, how and why GPs deprescribe in frail oldest-old patients with multimorbidity and polypharmacy, and to identify factors that influenced their decision to deprescribe. | GPs (n=157). | ▲ Percentage of GPs willing to de-prescribe at least one medication in the case of frail older patients with cardiovascular disease and compared with frail older patients without cardiovascular disease.<br>▲ Reasons for deprescribing<br>▲ Importance ratings given to factors influencing decision to deprescribe. |
| van Summeren et al, 2017[46] | Netherlands—general practice centres | Quantitative: Cross-sectional and implementation study. | To determine proposed and observed medication changes when using an outcome prioritisation tool during a medication review in older patients with multimorbidity and polypharmacy. A secondary aim was to explore the relationship between the prioritised health outcome of patients and the type of medication change, such as a stop, a dose adjustment, or a switch. | Patients aged 69 or over with two or more chronic diseases (one of which had to be cardiovascular disease) and daily use of five or more medications. (n=59)<br>GPs (n=17). | >Patients' priority rankings of the four health outcomes in the outcome prioritisation tool:<br>▲ Maintaining independence<br>▲ Remaining alive<br>▲ Reducing other symptoms<br>▲ Reducing pain<br>>Medication changes proposed by the GP, and observed in the patient records following incorporation of the priority rankings given by patients, into a medication review consultation. |

Continued

**Table 1** Continued

| Author and year of publication | Setting | Study type | Study aims | Target group and no of participants (n) | Outcomes measured |
|---|---|---|---|---|---|
| van Summeren et al, 2016[45] | Netherlands—general practice centres | Mixed-methods: Cross-sectional survey pilot and qualitative interviews to assess acceptability (semistructured and in-depth). | To explore whether an outcome prioritisation tool is appropriate in the context of medication review in family practice, focusing on its acceptability and practicality. | Patients aged 69 or over with two or more chronic diseases (one of which had to be cardiovascular disease) and daily use of five or more medications (n=60) GPs (n=13). | >Patients' prioritisation of the four domains of the outcome prioritisation tool:<br>▲ Maintaining independence<br>▲ Remaining alive<br>▲ Reducing other symptoms<br>▲ Reducing pain<br>>Family practitioners views on the acceptability and practicality of using the outcome prioritisation tool for medication review. |
| **Problem-based priorities** | | | | | |
| Junius-Walker et al, 2012[52] | Germany—general practice centres | Quantitative: RCT | To investigate whether a structured priority-setting consultation reconciles the often-differing doctor–patient views on the importance of problems. | Patients aged 70 or over (n=317) GPs (n=40). | ▲ Baseline importance rankings given by patients and clinicians to a list of problems generated from a geriatric assessment for each patient.<br>▲ Importance rankings given again after a structured consultation incorporating the baseline problem list and importance rankings and degree of reconciliation in doctor–patient agreement after the structured consultation. |
| Junius-Walker et al, 2011[53] | Germany—general practice centres | Quantitative: Cross-sectional Survey. | To gain insight into setting individual priorities with older patients using a priority definition that was coherent to the patients' life and doctors' work context. | Patients aged 70 or over and living at home (n=123) GPs (n=11). | Importance rankings given by patients and clinicians to a list of problems generated from a geriatric assessment for each patient. |
| Voigt et al, 2010[54] | Germany—general practice centres | Quantitative: Cross-sectional Survey. | To ascertain health priorities of older patients and treatment priorities of their GPs on the basis of a geriatric assessment and to determine the agreement between these priorities. | Patients aged 70 or over and at least one contact with the GP in the preceding 3 months (n=35) GPs (n=9). | ▲ Importance rankings given to problems generated from a geriatric assessment by patients and clinicians<br>▲ Degree of agreement between patients and clinicians on the above. |
| **Condition-focused priorities** | | | | | |
| Moore et al, 2014[36] | Canada—databases of all practising nurse practitioners, family practitioners and geriatricians in Ontario | Quantitative: Cross-sectional Survey. | To quantify how family physicians, nurse practitioners and geriatricians prioritise syndromes, diseases and conditions when caring for seniors. | Nurse practitioners (n=68) Family practitioners (n=84) Geriatricians (n=27). | Frequency and importance rankings given by family practitioners, nurse practitioners and geriatricians to 41 health issues known to arise in elderly patients |

Continued

**Table 1** Continued

| Author and year of publication | Setting | Study type | Study aims | Target group and no of participants (n) | Outcomes measured |
|---|---|---|---|---|---|
| Zulman et al, 2010[44] | USA—scheduled primary care visit for patients at nine veteran affairs facilities | Quantitative: Prospective Cohort Study. | To understand patterns of patient–provider concordance in the prioritisation of health conditions in patients with multimorbidity. | Patients with diabetes and hypertension who had their primary diabetes care provider enrolled in the study (n=1169) Primary care providers that is, physicians, physician assistants or nurse practitioners (n=92). | ▲ Patient rankings given in terms of their most important health concerns and providers rankings in terms of conditions most likely to affect each patient's outcomes ▲ Concordance between the importance ratings of patient-provider 'pairs'. |
| Herzig et al, 2019[56] | Switzerland—primary data were from 'Multimorbidity in Family medicine' study.[75] Patients enrolled by GPs during scheduled consultations. | Quantitative: Cross-sectional Survey. | To describe FPs' medical priority ranking of conditions relative to their prevalence in patients with multimorbidity. | Patients suffering from at least 3 of 75 chronic conditions on a predefined list (based on the International classification of primary care 2 (n=888) GPs (n=100). | Importance rankings given by family practitioners to the list of chronic conditions that each patient had on the day of their inclusion in the study. |
| Déruaz-Luyet et al, 2018 [58] | Switzerland—primary data were from 'Multimorbidity in Family medicine' study.[75] Patients enrolled by GPs during scheduled consultations. | Quantitative: Cross-sectional Survey. | To evaluate whether GPs could identify the condition that their patients with multimorbidity considered most important. | Patients suffering from at least 3 of 75 chronic conditions on a predefined list (based on the International classification of primary care 2, and receiving follow-up from their GP for at least the preceding 6 months (n=572 for main analysis, 585 for sensitivity analysis) GPs (n=100). | Whether there is agreement between what patients considered to be their most important health condition and what GPs thought patients considered to be their most important health condition. |
| **Treatment priorities** | | | | | |
| Caughey et al, 2017[47] | Australia—multidisciplinary ambulatory consulting service clinics at tertiary teaching hospitals | Mixed-methods: Structured quantitative interviews with patients then semistructured qualitative interviews with patients and clinicians. | To investigate how older patients with multimorbidity balance the benefits and harms associated with medication for prevention of CVD, and in the presence of competing health outcomes. To investigate the factors that clinicians consider when making treatment decisions for older patients with multimorbidity. | Patients aged 65 or older with 2 or more chronic conditions (n=15) Clinicians (n=5). | ▲ Patient willingness to take a medication when presented with different scenarios with variable degree of benefit, impact on daily living, adverse outcomes and impact on other comorbid conditions ▲ Patient-reported data during semistructured interviews where they were asked about their treatment preferences, medication effects and shared decision making ▲ Clinician reported data during semistructured interviews on treatment decisions, patient preferences and polypharmacy. |
| **Qualitative** | | | | | |

Continued

**Table 1** Continued

| Author and year of publication | Setting | Study type | Study aims | Target group and no of participants (n) | Outcomes measured |
|---|---|---|---|---|---|
| Kuluski et al, 2013 [37] | Canada—A Family Health Team in Ontario | Qualitative: semistructured interviews | To examine patient goals of care from the perspectives of older persons with multimorbidities, their family physicians and informal caregivers (ie, family member or friend who provides ongoing support) and then examine the extent of alignment between these three perspectives. | Patients aged 65 or older with a diagnosis of at least two chronic health conditions (n=28) Informal Caregivers of included patients (n=28) Family physicians (n=4). | >Patient, caregiver and physician reported data on goals of care for the patients >Degree of alignment of goals of care across patient, caregiver and physician 'triads' |
| Schoenberg et al, 2009 [38] | USA—senior centres, low-income senior housing complexes, churches and a civic meeting hall | Qualitative: in-depth interviews | To understand how vulnerable older adults with multimorbidity prioritise and manage their chronic conditions. | Patients aged 55 or older with a diagnosis of at least two chronic illnesses, from low-income backgrounds (n=41). | Patient-reported data from in-depth interviews, regarding their medical history, self-care procedures, patient prioritisation by means of health-related areas of worry and health-related 'expenditures' in terms of money, time and need for reliance on others. |
| Fried et al, 2008 [41] | USA—senior centres, doctors' practices and a congregate housing site | Qualitative: focus groups | To examine the ways in which older persons with multiple conditions think about potentially competing outcomes, in order to gain insight into how processes to elicit values regarding these outcomes can be grounded in the patient's perspective. | Patients aged 65 or older and were taking 5 or more medications (participants also had a minimum of 3 chronic conditions). | Patient-reported data regarding their perceptions of the interactions between their different illnesses and treatment regimens, goals of treatment and decisions regarding treatment. |
| Naik et al, 2016 [42] | USA—qualitative data from the VETCARES study,[76] in which participants recruited from the VA tumour registry | Qualitative: open-ended questions as part of mixed-methods interviews which also included structured questions. | To identify a taxonomy of health-related values that frame goals of care of older adults with multimorbidity who recently faced cancer diagnosis and treatment. | Veterans with a diagnosis of head and neck, gastric, oesophageal, or colorectal cancer, and diagnosis fell 1 month prior to the study's opening eligibility window (6 months) (n=146). | Patient-reported data regarding their priorities or concerns regarding their future healthcare decisions |
| Elliott et al, 2007 [43] | USA—Harvard Pilgrim Health Centre, a health maintenance organisation in New England | Qualitative: semistructured interviews. | To explore how older adults with multiple illnesses make choices about medicines. | Patients taking more than three medicines with purposive sampling to reflect symptomatic comorbidities and asymptomatic comorbidities and mental health issues (participants had a minimum of 3 comorbidities) (n=20). | Patient-reported data regarding beliefs about medicines, medicine-taking behaviour, historical versus potential choices between different medicines, and factors influencing these choices. |

Continued

**Table 1** Continued

| Author and year of publication | Setting | Study type | Study aims | Target group and no of participants (n) | Outcomes measured |
|---|---|---|---|---|---|
| Turner et al, 2016[48] | Australia—long-term care facilities in South Australia | Qualitative: nominal group technique. | To use nominal group technique to generate then rank factors that general medical practitioners, nurses, pharmacists and residents or their representatives perceive are most important when deciding whether or not to de-prescribe medication. | Residents/representatives of residents (n=11) GPs (n=19) Nurses (n=12) Pharmacists (n=14). | ▶ Generated factors important for deprescribing according to residents/resident representatives, GPs, nurses and pharmacists ▶ Priority rankings given by groups containing representatives from all of the above, to the list of priorities generated previously. |
| Lindsay, 2009[49] | UK—participants recruited from CHD registries in Greater Manchester as part of a larger RCT[77] | Qualitative: focus groups and two interviews. | To use the concepts of 'chronic illness trajectory' and 'biographical disruption' to examine how patients self-manage multiple chronic conditions and especially how they prioritise their conditions. | Participants from the parent study who had more than one chronic condition (ie, at least two) (n=53). | Patient-reported data regarding how they prioritised their multiple conditions, what strategies they used to cope with their conditions and barriers in being able to manage their illnesses. |
| Cheraghi-Sohi et al, 2013[50] | UK—secondary analysis of qualitative data from four other studies[78–81] | Qualitative: in-depth interviews. | To explore how and why people with multimorbidity prioritise some long-term conditions over others and what the potential implications may be for self-management activity, and in turn, suggest how such information may help clinicians negotiate the management of multimorbidity patients. | Participants from original studies who had two or more long-term conditions, and had given data regarding prioritisation (n=41). | Patient-reported data pertaining to prioritisation of their long-term conditions. |
| Morris et al[51] | UK—general Practices in North-West England | Qualitative: semistructured interviews. | To examine what influences self-management priorities for individuals with multiple long-term conditions and how this changes over time. | Patients with more than one chronic condition and at least one of Chronic Obstructive Pulmonary Disease, Irritable Bowel Syndrome or Diabetes (n=21). | Patient-reported data on management strategies and experiences with primary healthcare, and data from follow-up interviews on any changes in their illness management. |
| Hansen et al, 2015[55] | Germany—participants recruited from the 'Multicare cohort study'[82] | Qualitative: Focus groups | To identify reasons for disagreement regarding illnesses between patients and their GPs. | Patients who had 3 or more chronic conditions from a list of 29 conditions (n=21). GPs of the recruited patients (n=15). | Data from separate focus groups for patients and clinicians in which any communication problems and reasons for disagreement between patients and clinicians were explored. |

CVD, cardiovascular disease; GP, general practitioner; RCT, randomised controlled trial.

**Table 2** Summary of most important rankings for studies using the outcome prioritisation tool

| Study | Health outcome prioritisation as a tool for decision making among older persons with multiple chronic conditions[39] (%) | Health outcome prioritisation to elicit preferences of older persons with multiple health conditions[40] (%) | Outcome prioritisation tool for medication review in older patients with multimorbidity: a pilot study in general practice[46] (%) | Eliciting Preferences of multimorbid elderly adults in family practice using an outcome prioritisation tool[45] (%) | Aggregate ranking as most important (%) |
|---|---|---|---|---|---|
| Maintaining independence | 270 (75.6) | 34 (42.0) | 7 (36.8) | 19 (35.8) | 330 (64.7) |
| Staying alive | 40 (11.2) | 22 (27.2) | 6 (31.6) | 18 (34.0) | 86 (16.9) |
| Pain relief | 26 (7.3) | 17 (21.0) | 1 (5.3) | 6 (11.3) | 50 (9.8) |
| Symptom relief | 21 (5.9) | 8 (9.8) | 5 (26.3) | 10 (18.9) | 44 (8.6) |
| Total no of participants | 357 | 81 | 19* | 53 | 510 |

*Although there were 59 patients included in this study[46] priorities were only reported for 19 patients.

priorities, one study reported that 98% of a sample of 157 general practitioners (GPs) identified the 'quality of life for the patient', and 96% identified the 'life expectancy of the patient', as the most important factors in influencing their clinical decision making to deprescribe for elderly, patients with multimorbidity.[57]

### Priorities based on health problems

Three studies reported patient and GPs' priorities based on various health problems, following a geriatric assessment.[52–54] These problems were then categorised into domains, and the importance rankings for each of the domains were presented. Problems in the domains of 'social' 'mood' and 'function' recurrently featured in the top four of the most highly ranked priorities by patients across all three studies. In terms of the importance rankings by clinicians, problems in the domains of 'mood' and 'function' also featured in the top four importance rankings across all three studies, while 'social' problems were rated highly in one study[53] and problems in the domain of 'medication' were ranked highly in the other two studies.[52 54] Interestingly, the authors in one study[53] found that patients feeling 'emotionally affected' was the strongest predictor for a problem being rated as important (OR 11.1, 95% CI 6.73 to 18.33), whereas 'poor prognosis' was the strongest predictor for clinicians (OR 6.39, 95% CI 4.61 to 8.87)

### Condition-focused priorities

Two studies reported patient priorities in relation to specific conditions or groups of conditions,[44 58] in the context of multimorbidity. Zulman et al reported that 'diabetes/glycaemic control' was most frequently ranked as 'most important', with 'hypertension' coming second.[44] However, the sample of patients included in this study were all diabetic, hypertensive patients. Déruaz-Luyet et al found that musculoskeletal conditions, including back pain, were most frequently reported to be the most important conditions for their patients, however,

endocrine/metabolic conditions (including obesity) were second and cardiovascular conditions were third.[58]

Three studies reported condition-focused priorities of clinicians in the context of multimorbidity. Herzig et al reported the priorities of GPs alone,[56] and found that 'multiple sclerosis', 'mental retardation' and 'bronchus lung neoplasm' were all highly prioritised by their participants. Zulman et al reported the priorities of 'primary care providers' who consisted of physicians, physician assistants or nurse practitioners,[44] and found that diabetes was the top priority for primary care providers, with hypertension coming second, in alignment with their previously described patient priorities.[44] Moore et al examined the priorities of different types of clinicians, including family physicians, geriatricians and nurse practitioners,[36] and as with Zulman et al, found that diabetes was the top priority for family physicians and also nurse practitioners, whereas dementia was the top priority for geriatricians.[44] In addition, heart failure, atrial fibrillation and hypertension formed three of the top five conditions considered to be most important by the family practitioners in the study.[36]

### Treatment priorities

As part of a study to examine the influence of the risks and benefits of medications on treatment preferences of patients, Caughey et al also examined the priorities of patients in the face of 'competing outcomes'.[47] They found that 80% of participants would not be willing to take medication to reduce 'joint pain', if the medication increased their risk of a myocardial infarction by 10%. However, this was deduced from a sample of only 15 patients.[47]

### Agreement between patients and clinicians

Five of the included studies investigated the level of agreement in priority rankings between patients and their clinicians.[44 52–54 58] Three studies reported a low level of agreement between patient and clinicians' priority rankings.[52–54] Two of these studies used a Cohen's kappa

calculation to estimate the degree of agreement between the importance ratings of patients and clinicians, and the values of which were 0.18 and 0.11, respectively, indicating 'slight agreement' after allowing for chance.[53 54] One study used a weighted kappa calculation to measure the degree of agreement, which, at a preintervention point in this study, was low at 6%.[52]

Two studies reported that there was a 'high' level of agreement.[44 58] Déruaz-Luyet et al found that in the case of 54.9% (n=314) of their patients, the condition that their GP had considered to be either the first or second most important, was in the same disease group as the condition that the patient considered to be most important.[58]

Zulman et al reported that 60% of 'patient–provider pairs' had a 'high concordance', meaning that the same three conditions had been rated as top three priorities by both parties, or that two of the same conditions had been rated in the top three priorities by both parties.[44] In this case, given that the samples of patients were all diabetic and hypertensive could have led to a narrowing of the range of chronic diseases across the sample, which in turn could have led to an increased likelihood of agreement. However, the participant characteristics reported by the authors state that the patients had a mean of eight health conditions (SD 3.00), suggesting that the patients did not have a narrow range of chronic diseases. Furthermore, the questions posed to patients and providers were phrased differently, in that providers were asked to choose the top three most important medical concerns 'that are likely to affect health outcomes for this patient', whereas patients were asked to choose their top three most important health concerns. The authors acknowledge this in their paper, and justify this difference as being due to their aim of exploring the concordance in priorities about the 'most important problems facing the patient', rather which problems 'providers thought the patient would have prioritised', which, they argue, is a different concept to their aim.[44]

## QUALITATIVE SYNTHESIS

While our quantitative synthesis allowed us to investigate which health outcomes, diseases or treatments were important to patients with multimorbidity and their clinicians, our qualitative analysis enabled us to explore how prioritisation occurs. Below, we describe the key findings from our qualitative analysis.

### Mechanisms of prioritisation

In the qualitative studies that approached prioritisation from a disease-specific perspective, patients were able to identify an illness as their main priority.[49 50] For many patients, prioritisation appeared to be driven by their experience of the illness, which formed part of its 'meaning as consequence[50]' as phrased by Cheraghi-Sohi et al. The 'consequences' of an illness consisted of the impact that the illness was having on the patients' everyday lives, which included functional limitation and the symptomatic burden of the illness, including its 'unpredictability' (table 3).[49] For others, prioritisation appeared to be driven by their perception of the risk now and in the future with respect to functional deterioration and mortality.

In other studies, patients framed their priorities between quality of life versus length of life (table 3).[42] Patients in the study by Naik et al who were adults with multimorbidity and suffering from cancer, prioritised 'quality of life' more highly than 'length of life'.[42] This was also reflected in the findings of Fried et al, who found that when considering medication with competing outcomes in terms of extending life compared with quality of life, participants appeared to prioritise preserving quality of life.[41]

van Summeren et al found that prioritisation was 'difficult' when there was no 'specific need' for a treatment decision to be made.[45] This concept of a difference in prioritisation based on hypothetical, or experiential levels, was also shared in the findings of Elliott et al[43] and Fried et al.[41]

Where clinicians' perspectives were explored alongside patients, clinicians reported that exploring patients' priorities was 'extremely important' when managing 'competing interests'[47] and beneficial in providing patient-centred care.[45] Some clinicians in the mixed-methods study carried out by van Summeren et al reported that exploring their patients' priorities allowed them to have a 'deeper understanding' of the patient, helped with making patient-centred treatment decisions and advance care planning (table 3).[45] However, other clinicians in the same study found exploring patient priorities to be

**Table 3** Examples from included studies for key concepts relating to mechanisms of prioritisation

| | Concept | Examples from included studies |
|---|---|---|
| Mechanisms of prioritisation | Unpredictability of symptoms | 'My final issue is diverticulitis. In many ways that is the thing that makes the most impact on my life because of the unreliability of it. You make plans to do something to go somewhere and at the last minute you don't dare leave the house because you don't leave the loo. In itself it's not an important medical issue. It's the social problem more than anything else.' Lindsay et al[49] |
| | Quality of life versus length of life | 'If you don't feel good, you can't take care of yourself and you have to depend on somebody else, what's the good of living another 10 years?' Fried et al[41] |
| | Facilitating clinicians' decision making | 'In future, I'll be happier to be more decisive in keeping an eye on what we do and do not do as regards this patient.' Van Summeren et al[45] |

**Table 4** Examples from included studies for key concepts relating to factors influencing prioritisation

| | Concept | Examples from included studies |
|---|---|---|
| **Factors influencing prioritisation** | Functional ability | 'I mean, because I have to be mobile, I am living on my own, no one is going to take care of me, I have got to look after myself…' Cheraghi-Sohi *et al*[50] |
| | Mortality | 'Well I really do worry the most about the high blood pressure. 'Cause see you know you got arthritis and you can tell when it's coming on. But you can't hardly tell about high blood pressure. It can just hit you like that [snaps fingers] ….' Lindsay *et al*[49] |
| | Symptom control | 'I would not want to live with pain. I won't allow that to happen'Naik *et al*[42] |
| | Disparity in prioritisation of symptom control | '… I talk [to her] for a quarter of an hour about this and that every time after which she replies, 'but my vertigo,' and I answer every time, well, unfortunately there is nothing I can do about it, we have already tried and done everything. But it is probably the first diagnosis she will mention: 'What are you suffering from?'. 'Vertigo'. For me, this would be somewhere all the way at the bottom.' Hansen *et al*[55] |
| | Treatment burden | 'It's the knee that's the most concerning because everything else is controlled by tablets. The knee is a problem because if I have one little slip I'm in plaster again for 6 weeks.' Lindsay *et al*[49] |

difficult due its 'novelty' and the fact that it represented a change to their usual consultations.[45]

### Factors influencing prioritisation

Our analysis revealed that there were a number of factors that appeared to influence how both patients and clinicians arrived at their priorities, and which priorities they chose.

#### Functional ability

Preserving functional ability as a priority for patients was a dominant concept across the majority of the qualitative studies.[37 38 41 42 47 49 51] Preserving independence emerged as the most significant reason for prioritising functional ability for patients, and maintaining the ability to engage in activities of daily living, mobility, maintaining cognitive ability and wanting to avoid being a 'burden' or lacking social support to help them cope with functional deterioration (table 4).[38 49 50]

Conditions, which caused limitation to patients' ability to self-manage their health conditions, led to a 'tension' between the patients' expectations of themselves and what they were physically able to do.[51] Lifestyle management, particularly reduced ability to exercise and the adverse impact of this on weight, was cited as part of patients' ability to self-manage.[49]

Maintaining patients' functional ability was reported as a priority by some clinicians.[37 47] Clinicians considered the wider implications of the patients' functional deterioration, particularly cognitive deterioration, and spoke of wanting to reduce the risk of 'burnout' for the patients' family members/caregivers.[37]

#### Mortality

Reducing the risk of mortality emerged as a recurrent priority for clinicians.[47 55] Caughey *et al* found that clinicians prioritised mortality in younger (less than 65 years) patients with multimorbidity rather than older patients with multimorbidity, as they felt they could be more 'aggressive' in their treatment.[47] Reducing the risk of mortality also emerged as a priority for patients across a number of studies.[37 38 42 43 50 51] Some patients found the asymptomatic nature of hypertension to be concerning; hence, the consequences of hypertension could be unpredictable, compared with some other chronic illnesses where symptoms can give warning of onset and severity (table 4).[38 43]

#### Symptom control

The symptomatic burden of a condition contributed to its 'meaning as consequence' for patients.[50] Symptoms were cited as being a cause of functional limitation,[38 49] and in some cases their 'unpredictability' could cause significant disruption to patients' daily lives.[49] Symptom control was reported to be a priority by some clinicians.[37 47] However, clinicians in one study considered symptom control to be less important, particularly when there was no risk of mortality.[55] In these cases, clinicians seemed to be aware that patients may still be prioritising symptom control highly, even if the clinicians did not (table 4).

#### Treatment burden

Factors related to the treatment burden of an illness appeared to adversely impact prioritisation for patients, leading to de prioritisation of certain medications and treatments.[38 41 43 48] Elliot *et al* reported that cost and distressing side effects were factors which led patients to stop taking a medication.[43] Similarly, Fried *et al* found that patients reported unpleasant side effects to be a 'competing outcome', which negatively influenced their decision regarding continuing a medication.[41] However, difficulty with achieving control over the management of an illness, as well as requirement for high levels of engagement with self-management, emerged as factors that contributed to the prioritisation of an illness by some patients (table 4).[49]

### DISCUSSION

Prioritisation as a concept is broad, context dependent and difficult to confine into a single definitive definition.

As a result, determining what can be interpreted as a health outcome or treatment priority as part of our study selection in this review was inherently difficult. We excluded some studies that investigated the preferences of patients with multimorbidity or clinicians, in contexts that we judged to be different to the aim of this review. These included patient preferences for healthcare delivery,[59 60] levels of engagement with self-management practices[61 62] and clinicians' experiences of the management of patients with multimorbidity.[18 63 64] While these studies represent very important areas of research, they were not within the scope of our aim in this review, that is, identifying studies that report the health outcome and treatment priorities of patients with multimorbidity or those of clinicians in relation to patients with multimorbidity. A discussion from our synthesis of findings of the included studies in this review is presented below.

### Health outcome and treatment priorities

From our findings, patients' prioritisation appeared to be driven by weighing up the empirical compared with the hypothetical impact of a disease, whereby the empirical impact of a disease, which included its impact on function, symptomatic and treatment burden, was the most dominant driver of prioritisation. This is consistent with the findings of previous literature showing patients with rheumatoid arthritis who had reported experiencing higher levels of pain, were more likely to report pain as a priority.[65]

Among empirical factors, preserving functionality emerged as most highly prioritised by patients among the quantitative studies that took a health outcome approach,[39 40 46] while 'function' was a domain that was prioritised highly by both patients and clinicians in the studies where prioritisation of various health problems were investigated.[52–54] From our qualitative findings, functional ability formed a key part of the preservation of various aspects of the patients' independence and their quality of life, as well as their ability to self-manage. Existing evidence shows that the prevalence of multimorbidity is highest in those aged over 65 years,[66] and the population for the majority of the included studies were older adults with multimorbidity. This could provide an explanation for why preserving functionality was highly prioritised.

Prioritisation was not a static process and was subject to change, based on factors such as illness exacerbations, life events, whether there was a need for a treatment decision to be made, and whether the priority related to retrospective or prospective healthcare.[49 51] When considering the hypothetical impact of an illness, perceptions of future risk came into play, and in particular, the risk of mortality.[43] This was particularly evident in relation to cardiovascular disease, where patients appeared to perceive the risk of mortality to be high.[38]

Risk of mortality was a dominant driver for prioritisation among clinicians. This was shown in our quantitative synthesis, where among studies assessing disease-specific priorities, conditions with a higher risk of mortality, such as cardiovascular disease and diabetes, recurrently emerged as being highly prioritised by clinicians[36 44 56] and differentiated by age.[47] This age-based consideration could explain why clinicians prioritised 'quality of life for the patient' as higher, although marginally, than 'life expectancy of the patient' in their clinical decision-making for deprescribing for elderly, patients with multimorbidity.[57]

Smith *et al* previously developed a 'Core Outcome Set'[67] in which a Delphi consensus panel formed of 26 international health experts, identified and prioritised a set of outcomes tailored for application to research studies targeting patients with multimorbidity. Mortality, mental health outcomes and quality of life featured most highly in their list of prioritised outcomes, which also emerged in this review. However, we found that relatively few studies reported the prioritisation of mental health outcomes, with the exception of the studies that took a problem-based approach to prioritisation, where problems with regard to 'Mood' were prioritised highly by both patients and clinicians.[52–54]

Our findings show a varying degree of agreement between the priorities of patients with multimorbidity and clinicians. Previous studies carried out in the context of diabetes,[68] and psoriasis[69] have found a low level of agreement on health outcome and treatment priorities between patients and clinicians, which correlates with the findings of some studies included in this review,[52–54] but not others.[44] The nature of the patients' illnesses emerged as a factor for concordance or discordance of priorities with their clinicians.[37] Patients and clinicians were in agreement in situations where patients were currently experiencing an exacerbation of a particular condition, or had a 'stable' state of health. However, in patients who suffered from illnesses with more complex courses, discordance of priorities tended to occur between patients and clinicians.[37]

In recent times, the traditional paternalistic model for the doctor–patient relationship has given way to an egalitarian model,[70] where doctors and patients each play an equitable role in a shared decision-making process, which places the patient at its core and thus achieving greater patient-centred care.[70 71] A shared agreement between patients and doctors on treatment priorities have been highlighted to play an important part in achieving patient-centred care and creating a therapeutic alliance, the benefits of which can include improved treatment adherence.[70 71] Indeed, Jowsey *et al* found that agreement between patients and clinicians in the formulation of care plans promoted adherence to these plans, whereas a lack of agreement led to disengagement with care plans by patients.[72]

### Strengths and limitations

To our knowledge, this is the first systematic review to assimilate and compare the findings of existing literature on the health outcome and treatment priorities of both patients and clinicians for patients living with multimorbidities. In this review, we have been able to add a novel line of argument to the ongoing discussion on this

subject. By incorporating papers using both qualitative and quantitative methodologies, we have been able to explore patterns and relationships in the findings of a wide range of studies, thus creating a comprehensive and well-rounded systematic review.

There are noteworthy limitations. We did not include the term 'comorbidity', in our search terms, and while 'comorbidity' is distinctive from multimorbidity, there is also some conceptual overlap between the two terms. We felt that including 'comorbidity' in our search strategy would identify studies focusing on a specific condition rather than multimorbidity.

A number of the quantitative studies did not use prevalidated tools to ascertain priorities,[36 54 56–58] leading to a risk of measurement bias, which could limit the generalisability of findings in this review. All of the included studies were conducted in developed, western countries, which limits the global generalisability of our findings, as the priorities of patients with multimorbidity and of clinicians in developing and/or eastern countries may differ to the findings of this review.

We also detected a large variation in the tools used to ascertain priorities, which meant that carrying out a meta-analysis to synthesise the findings of the quantitative studies was not possible. Yet, we have tried to mitigate the lack of meta-analysis by using a well-described and well-established method of narrative synthesis,[33] in order to maintain rigour and transparency.

Another limitation is that, in our inclusion criteria, we chose to also include studies which did not explicitly specify a definition of multimorbidity as 'two or more chronic conditions' in their inclusion criteria but had a sample of participants that were reflective of multimorbidity (ie, with a minimum of two chronic conditions which could be identified from participant demographic data). We chose to do this as in the absence of a universally accepted and uniform definition of multimorbidity, we sought to base our judgement on the inclusivity of each paper on its value in answering our review question. This, along with the previously discussed difficulty in defining prioritisation, may have introduced a degree of subjective interpretation in the process of study selection, despite our attempt to mitigate this by incorporating independent review of the results of our literature searching by two reviewers in duplicate.

### Recommendations for the future

We recommend that future guidelines developed for clinicians in the management of multimorbidity highlight the need to elicit and consider both short-term and long-term priorities for their patients', as our review has shown that patients' priorities for their current illness experiences and future risks posed by illnesses, may differ. In accordance with current National Institute for Health and Care Excellence guidance, we also reiterate the need to review these priorities continually, and particularly when exacerbations, changes to illness course or treatment regimens,

or other wider socially contextualised changes occur in their patients' lives.

There was a large variation in how priorities were ascertained, and in the tools used to ascertain priorities. The relative lack of standardised and validated tools for use to ascertain patient priorities in everyday clinical practice has also been described in previous literature.[73] We highlight a need for the development of a standardised and validated tool that is acceptable to both patients and clinicians, and can be used to ascertain patient priorities in the multiple dimensions described in this review. Such a tool would a valuable aid to treatment decision making, advance care planning and achieving patient-centredness for patients living with multimorbidity.

### CONCLUSION

The findings from this review show the priorities of patients and clinicians can have varying degrees of concordance, being mostly low,[52 54] in alignment with previous findings in single disease contexts.[68 69] We have found that the mechanisms of prioritisation can also differ between our two groups, in that patients are driven by illness experiences, whereas clinicians may be focused on managing longer term risks. Understanding these differences can help clinicians to better recognise situations where the patients' priorities may be different to theirs and elicit the most important priorities for their patients.

**Contributors** HS: design of research question and methodology, data searching, data extraction, data analysis and manuscript development; MS: design of methodology, data extraction, data analysis and manuscript development; RF: data searching and data extraction; CG: data analysis and manuscript development; UK: data analysis and manuscript development; MJD: data analysis and manuscript development; KK: design of research question and methodology, manuscript development; SS: conception of the idea for this review, design of research question and methodology, data extraction, data analysis and manuscript development. All authors have approved the final manuscript.

**Funding** HS is funded by the NIHR academic clinical fellowship award.

**Competing interests** None declared.

**Patient consent for publication** Not required.

**Provenance and peer review** Not commissioned; externally peer reviewed.

**Data availability statement** All data relevant to the study are included in the article or uploaded as supplementary information.

**ORCID iDs**
Harini Sathanapally http://orcid.org/0000-0001-8283-1411
Kamlesh Khunti http://orcid.org/0000-0003-2343-7099

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
