## [Reviewer comments · BMJ Open]

ARTICLE DETAILS

TITLE (PROVISIONAL)	Priorities of patients with multimorbidity and of clinicians regarding treatment and health outcomes: a systematic mixed studies review
AUTHORS	Sathanapally, Harini; Sidhu, Manbinder S.; Fahami, Radia; Gillies, Clare; Kadam, Umesh; Davies, Melanie; Khunti, Kamlesh; Seidu, Samuel

VERSION 1 – REVIEW

REVIEWER	JOSE M. ABAD DIEZ HEALTH CARE DEPARTMENT REGIONAL GOVERNMENT OF ARAGON SPAIN
REVIEW RETURNED	16-Aug-2019

GENERAL COMMENTS	There are some errors regarding the identification of the different studies included in the review that must should be revised. For example, in page 10 of the manuscript, regarding the quality assessment of the qualitative studies, the authors identify three studies with only limited description of the analytic process, whereas they present only two referenes (48 and 49). And they refer to two studies not presenting any quotes, and it seems that there is an erro in the reference (should be 46 and 49, instead of 48 and 49).
--

REVIEWER	Tanisha Jowsey Centre for Medical and Health Sciences Education, University of Auckland, New Zealand
REVIEW RETURNED	17-Sep-2019

GENERAL COMMENTS	Thank you for the opportunity to review this paper. Seldom have I encountered a literature review so well written as this. The methods are well detailed. I was particularly interested in the qualitative findings. van Summeren et al suggested that clinicians exploring patient priorities was considered by clinicians as novel and difficult. I think the patient priority and person-centred care education is now coming through very strongly in medical schools the world over. It would be interesting if their study were to be conducted again in 15 years to see if the cultural shift has occurred in healthcare. I like the way the findings are presented in Tables 3 and 4. In the discussion I was expecting the focus to be less on what you found and more on the wider picture of patient-centred/person-centred care. I guess I was looking for guidance on what the implications of competing patient/professional priorities are for each. That is, what does it mean for patients in terms of the decisions they make each day to manage their lives in the context of multiple
---

	illnesses? What does it mean for professionals in the advice they offer to patients? What are the implications for multi-morbid care plans? (I actually have a paper on that, suggesting patient priorities must be factored in to care plans and must speak the language of patients or else they are a bit of a waste of time Jowsey et al. Sociology of Health & Illness Vol. 38 No. 6 2016). There is a wealth of literature on patient-doctor relationships informing health experiences and outcomes. This goes back to Parsons in 1963. Actually, we might argue it goes back to Hippocrates. In any case, my view is that if you were to add another small paragraph in your discussion about these wider issues your paper would be even better.
--	--

REVIEWER	Susan Smith RCSI Ireland
REVIEW RETURNED	01-Oct-2019

GENERAL COMMENTS	Thank you for asking me to review this paper reporting a mixed methods systematic review of treatment and health outcome prioritisation of patients with multimorbidity and clinicians. The authors are to be commended for undertaking this very ambitious piece of work combining both quantitated and qualitative syntheses. It addresses an important topic for patients, clinicians and health care systems. Positives include the quality assessment and use of appropriate reporting guidelines. However as currently reported there are major limitations. These include:  • There is lack of clarity around the exact meaning of prioritization and some of the included studies seem to use different definitions. Prioritization could mean the symptoms or issues that a patient finds most important to address during a consultation, and this type of prioritization is recommended in all of the existing multimorbidity clinical guidelines. These guidelines are not referred to in the paper which is a limitation. It is hard to see how a clinician could know these in advance or in general and this kind of prioritization would really only work retrospectively – did the clinician recognize what the patient’s priority was. • However prioritization could be more condition focused and also relate to which of the patient’s conditions they thinks is most important and should be focused on. This could more easily be compared to clinician rankings and this in turn could be compared to the prioritise Asian or ranking of the patients condition by their treating clinician • From the perspective of clinicians, prioritization can also refer to the supports they need to deal with complex multimorbidity and there is existing qualitative literature on this which is not considered • The aim focuses very much on identifying prioritization and comparing patient and clinicians but one of the most interesting part of the results is the qualitative analysis of the prioritization processes. This seems to me to be the most novel and useful part of the review given the clinical recommendations to address patient priorities • The search string is provided and as the authors acknowledge they omit some important related terms such as comorbidity. While I agree with this approach in terms of inclusion, the MeSH term for multimorbidity is relatively recent and some MM studies may only be identified using the older Co-M term. They then go on to include some co-morbidity studies picked up by the MM search despite this justification The number of abstracts retrieved by the search seems quite small and given the broad topic. For example the Cochrane
---

	Multimorbidity review search retrieves thousands more titles. It is a very challenging area to do systematic searches but I was surprised to see some well-known qualitative papers examining patients experiences of multimorbidity missing eg work of P Noel et al. These papers may have been missed as they are not specifically focused on prioritization. However they do report issues that patients with multimorbidity prioritise particularly function and managing medicines.  • In relation to health outcomes, the authors could consider the related core outcome set in multimorbidity, published in the Annals Family Medicine as this was also an attempt to prioritise outcomes that matter to key stakeholders including patients and clinicians. Interestingly the core outcome set ranks mental health outcomes as being very important whereas these don't seem to feature in the current paper, which is surprising. • The results don't clearly report triangulation of quantitative and qualitative findings • Specific queries relating to included studies may relate to reporting and include:  o Some of the studies actually seem to have participatns with just one condition who may also have another risk factor but this is not necessarily in keeping with the inclusion criteria eg Fried 2011 and 2006 o Zulman et al seem to ask patients and clinicians different questions so hard to se how can be directly compared o Junius-Walker and Voigt – no patient inclusion criteria reported other than age. If presuming MM on basis of age that would likely bring in a lot of other studies o Mantelli et al has a clear focus on clinician's attitude to deprescribing but what is this is not a patient priority? o Caughey et al examine preferences int he context of scenarios which again is different to actual priorities presenting in clinical practice o Schoenberg – description of the outcome does not seem congruent with the study aims o Naik et al- I think this is a very focused population with named specific conditions and not MM o Elliot et al: outcomes seem to be focused on medicines o Cheraghi Sohi seems to be another qualitative synthesis and woud it be more appropriate to screen included studies and see which were eligible rather than having two layers of interpretation • Table 1 – might be easier to group similar study types together and address the different types of prioritization more clearly this way • Results on disease-specific priorities do not seem as relevant to a review focusing on multimorbidity • The discussion is relatively brief and I think could address some of the issues I have raised above about definitions and process for prioritization • The recommendation to include prioritization in future guidelines reflects the author's lack of awareness of these guidelines. For example, NICE 2016, Ariadhne principles, Wallace BMJ and Christiane Muth's paper on overview of guidelines
--	--

VERSION 1 – AUTHOR RESPONSE

Reviewer: 1

Reviewer Name: JOSE M. ABAD DIEZ

Institution and Country: HEALTH CARE DEPARTMENT, REGIONAL GOVERNMENT OF ARAGON, SPAIN

Please state any competing interests or state 'None declared': NONE DECLARED

Please leave your comments for the authors below

- There are some errors regarding the identification of the different studies included in the review that must should be revised. For example, in page 10 of the manuscript, regarding the quality assessment of the qualitative studies, the authors identify three studies with only limited description of the analytic process, whereas they present only two referenes (48 and 49). And they refer to two studies not presenting any quotes, and it seems that there is an erro in the reference (should be 46 and 49, instead of 48 and 49).

Response: We thank the reviewer for noting these errors. We have corrected these by adding the missing third reference to the text relating to qualitative studies with only a limited description of the analytic process (page 10, line 17). We have also added a third reference (numbered 45, reference in the original draft was 46 however it is now 45 due to revisions that have been made), to the text regarding the studies not presenting any quotes (page 10, line 18), as there were three studies not presenting any quotes, rather than two as we had mistakenly stated in the original paper.

Reviewer: 2

Reviewer Name: Tanisha Jowsey

Institution and Country: Centre for Medical and Health Sciences Education, University of Auckland, New Zealand

Please state any competing interests or state 'None declared': None declared

Please leave your comments for the authors below

- Thank you for the opportunity to review this paper. Seldom have I encountered a literature review so well written as this. The methods are well detailed.

- I was particularly interested in the qualitative findings. van Summeren et al suggested that clinicians exploring patient priorities was considered by clinicians as novel and difficult. I think the patient priority and person-centred care education is now coming through very strongly in medical schools the world over. It would be interesting if their study were to be conducted again in 15 years to see if the cultural shift has occurred in healthcare.

- I like the way the findings are presented in Tables 3 and 4.

- In the discussion I was expecting the focus to be less on what you found and more on the wider picture of patient-centred/person-centred care. I guess I was looking for guidance on what the implications of competing patient/professional priorities are for each. That is, what does it mean for patients in terms of the decisions they make each day to manage their lives in the context of multiple illnesses? What does it mean for professionals in the advice they offer to patients? What are the implications for multi-morbid care plans? (I actually have a paper on that, suggesting patient priorities must be factored in to care plans and must speak the language of patients or else they are a bit of a waste of time Jowsey et al. *Sociology of Health & Illness* Vol. 38 No. 6 2016). There is a wealth of literature on patient-doctor relationships informing health experiences and outcomes. This goes back to Parsons in 1963. Actually, we might argue it goes back to Hippocrates. In any case, my view is that if you were to add another small paragraph in your discussion about these wider issues your paper would be even better.

Response: We thank the reviewer for their feedback. We used the published paper suggested [2] to re-frame the discussion to elaborate on patient-centred care and agreement between patients and doctors:

"In recent times, the traditional paternalistic model for the doctor-patient relationship has given way to an egalitarian model [3], where doctors and patients play an equal part in a shared-decision making process, which places the patient at its core and thus becomes patient-centred care [4][3]. A shared agreement between patients and doctors on treatment priorities have been highlighted to play an

important part in achieving patient-centred care and creating a therapeutic alliance, the benefits of which can include improved treatment adherence [3, 4]. Indeed, Jowsey et al found that agreement between patients and clinicians in the formulation of care plans promoted adherence to these plans, whereas a lack of agreement led to disengagement with care plans by patients [2].”

Reviewer: 3

Reviewer Name: Susan Smith

Institution and Country: RCSI, Ireland

Please state any competing interests or state 'None declared': None declared

Please leave your comments for the authors below

Thank you for asking me to review this paper reporting a mixed methods systematic review of treatment and health outcome prioritisation of patients with multimorbidity and clinicians. The authors are to be commended for undertaking this very ambitious piece of work combining both quantitated and qualitative syntheses. It addresses an important topic for patients, clinicians and health care systems. Positives include the quality assessment and use of appropriate reporting guidelines.

However as currently reported there are major limitations. These include:

- There is lack of clarity around the exact meaning of prioritization and some of the included studies seem to use different definitions. Prioritization could mean the symptoms or issues that a patient finds most important to address during a consultation, and this type of prioritization is recommended in all of the existing multimorbidity clinical guidelines. These guidelines are not referred to in the paper which is a limitation. It is hard to see how a clinician could know these in advance or in general and this kind of prioritization would really only work retrospectively – did the clinician recognize what the patient’s priority was.
- However prioritization could be more condition focused and also relate to which of the patient’s conditions they think is most important and should be focused on. This could more easily be compared to clinician rankings and this in turn could be compared to the prioritise Asian or ranking of the patients’ condition by their treating clinician
- From the perspective of clinicians, prioritization can also refer to the supports they need to deal with complex multimorbidity and there is existing qualitative literature on this which is not considered
- The aim focuses very much on identifying prioritization and comparing patient and clinicians but one of the most interesting part of the results is the qualitative analysis of the prioritization processes. This seems to me to be the most novel and useful part of the review given the clinical recommendations to address patient priorities
- The search string is provided and as the authors acknowledge they omit some important related terms such as comorbidity. While I agree with this approach in terms of inclusion, the MeSH term for multimorbidity is relatively recent and some MM studies may only be identified using the older Co-M term. They then go on to include some co-morbidity studies picked up by the MM search despite this justification. The number of abstracts retrieved by the search seems quite small and given the broad topic. For example the Cochrane Multimorbidity review search retrieves thousands more titles. It is a very challenging area to do systematic searches but I was surprised to see some well-known qualitative papers examining patients experiences of multimorbidity missing eg work of P Noel et al. These papers may have been missed as they are not specifically focused on prioritization. However they do report issues that patients with multimorbidity prioritise particularly function and managing medicines.

Response: Thank you, the paper “Collaborative care needs and preferences of primary care patients with multimorbidity” by P Noel et al did emerge in our literature search and did undergo a full text review. However, the decision was made to not include this paper in this review, as we felt the priorities and preferences explored in this study related to healthcare delivery rather than health outcome and treatment preferences which is the focus of our systematic review.

We attempted to keep our search criteria as wide as possible whilst still maintaining focus on the review questions and worked with our local medical librarians to achieve this. We agree this was a very challenging area in which to carry out a literature search and we accept there is an inevitable degree of subjectivity and judgement in the study selection. We attempted to mitigate this by incorporating independent review of the outcomes of our literature search by two reviewers, however we recognise this to be a limitation, and have made this clearer in our paper:

“...along with the previously discussed difficulty in defining prioritisation, may have introduced a degree of subjectivity in the process of study selection, despite our attempt to mitigate this by incorporating independent review of the results of our literature searching by two reviewers.”

Another difficulty was defining the boundaries and scope of this review, particularly with defining what we mean by prioritisation. We considered that prioritisation was context-dependent, and in this instance, chose to focus on health outcome and treatment priorities. Along with aforementioned paper by P Noel et al, our search did reveal other papers that investigated priorities and preferences in terms of healthcare organisation and service delivery, and after carrying out a review of these papers, we chose to exclude these on the basis that they did not address the questions that we set out for this review. We have amended our paper to discuss the difficulty with defining prioritisation and decisions we made in our study selection process, in the discussion section of our paper:

“Prioritisation as a concept is broad, context-dependent and difficult to confine into an objective definition. As a result, judging what “counts” as a health outcome or treatment priority as part of the study selection in this review, was inherently difficult. We excluded some studies that investigated the preferences of multi-morbid patients or clinicians, in contexts that we judged to be different to the aim of this review. These included patient preferences for healthcare delivery [5][6], levels of engagement with self-management practices [7][8] and clinicians’ experiences of the management of multi-morbid patients [9][10][11]. Whilst these studies represent very important areas of research, they were not within the scope of our aims in this review of identifying studies that report the health outcome and treatment priorities of multi-morbid patients or those of clinicians in relation to multi-morbid patients.”

- In relation to health outcomes, the authors could consider the related core outcome set in multi-morbidity, published in the Annals Family Medicine as this was also an attempt to prioritise outcomes that matter to key stakeholders including patients and clinicians. Interestingly the core outcome set ranks mental health outcomes as being very important whereas these don’t seem to feature in the current paper, which is surprising.

Response: Thank you, we have amended our discussion section to include a statement on how our findings relate to the findings of the COSmm paper published in the Annals of Family medicine: “Smith et al previously developed a “Core Outcome Set” [12] in which a Delphi consensus panel formed of 26 experts from across the globe, identified and prioritised a set of outcomes tailored for application to research studies targeting multi-morbid patients. Mortality, mental health outcomes and quality of life featured most highly in their list of prioritised outcomes, which also emerged in this review. However, we found that relatively few studies reported the prioritisation of mental health outcomes, with the exception of the studies that took a problem-based approach to prioritisation, where problems in the domain of “Mood” were prioritised highly by both patients and clinicians [13-15].”

- The results don’t clearly report triangulation of quantitative and qualitative findings.

Response: Thank you, we sought to synthesise the qualitative and quantitative data separately in order to maintain rigour in our analytic process. From the results of our two separate syntheses, we considered how the findings of the two syntheses may relate to each other, particularly where our qualitative findings may provide possible explanations for our quantitative findings. The outcome of this is described in the discussion section. We apologise that we did not make this clear in the paper and have now endeavoured to do so:

“From the results of our narrative synthesis of the quantitative studies and meta-ethnography of the qualitative studies, we considered how the findings of the two syntheses may relate to one another, particularly where our qualitative findings may provide possible explanations for our quantitative findings. The outcome of this process is described in the discussion section.”

- Specific queries relating to included studies may relate to reporting and include:
 - Some of the studies actually seem to have participants with just one condition who may also have another risk factor but this is not necessarily in keeping with the inclusion criteria eg Fried 2011 and 2006-

Response: Thank you, in our inclusion criteria we chose to also include studies which did not specify a definition of multi-morbidity as “two or more chronic conditions” in their inclusion criteria but had a sample of participants that were reflective of multi-morbidity (i.e. with a minimum of two chronic conditions). We chose to do this as in the absence of a universally accepted and uniform definition of multi-morbidity, we sought to base our judgement on the inclusivity of each paper on its value in answering our review question. However, we recognise that this has introduced an element of subjectivity into this review and that we did not acknowledge this as a limitation in our original paper, we have made amendments to make this limitation more explicit.

“Another limitation is that in our inclusion criteria we chose to also include studies which did not specify a definition of multi-morbidity as “two or more chronic conditions” in their inclusion criteria but had a sample of participants that were reflective of multi-morbidity (i.e. with a minimum of two chronic conditions). We chose to do this as in the absence of a universally accepted and uniform definition of multi-morbidity, we sought to base our judgement on the inclusivity of each paper on its value in answering our review question. This, along with the previously discussed difficulty in defining prioritisation, may have introduced a degree of subjectivity in the process of study selection, despite our attempt to mitigate this by incorporating independent review of the results of our literature searching by two reviewers.”

Where a study did not specify a definition of multi-morbidity in their inclusion criteria, we looked at the participant characteristics to judge whether the sample of participants were likely to be suffering from at least two chronic conditions. In the case of the 2011 study carried out by Fried et al, the participants had a mean of 2.9 chronic conditions, which we judged to be fulfilling our criteria for multi-morbidity. In the case of the 2006 study carried out by Fried et al, we felt the participant characteristics were reflective of multi-morbidity i.e. rate of hospitalisation, however we can see that compared to all of the other studies that did not specify a definition of multi-morbidity, we cannot deduce that the participants were likely to be suffering from at least two chronic conditions. We made efforts to contact the authors of this paper but did not receive a response. Therefore, we have removed this paper from our results, and amended the remainder of the paper accordingly.

- Zulman et al seem to ask patients and clinicians different questions so hard to see how can be directly compared-

Response: We agree that in this study the providers being asked to consider the top three most important medical concerns “that are likely to affect health outcomes for this patient” is asking a different question to the one that had been posed to the patients in this study. The authors acknowledge this in their paper, and justify this difference as being due to their aim of exploring the concordance in priorities about the “most important problems facing the patient”, rather which problems “providers thought the patient would have prioritised”, which, they argue, is a different concept to their aim. We have amended the results section of our paper to include this point in our description of this study.

“Furthermore, the questions posed to patients and providers were phrased differently, in that providers were asked to choose the top three most important medical concerns “that are likely to affect health outcomes for this patient”, whereas patients were asked to choose their top three most

important health concerns. The authors acknowledge this in their paper, and justify this difference as being due to their aim of exploring the concordance in priorities about the “most important problems facing the patient”, rather which problems “providers thought the patient would have prioritised”, which, they argue, is a different concept to their aim [16].”

- Junius-Walker and Voigt – no patient inclusion criteria reported other than age. If presuming MM on basis of age that would likely bring in a lot of other studies-

Response: Thank you, as discussed above, we chose to also include studies which did not specify a definition of multi-morbidity as “two or more chronic conditions” in their inclusion criteria but had a sample of participants that were reflective of multi-morbidity (i.e. with a minimum of two chronic conditions). In the case of both the studies carried out by Junius-Walker et al in 2011 and 2012, the mean number of health problems presented in the baseline characteristics tables in each paper were more than two. As discussed earlier, in our inclusion criteria we sought to also include papers that did not specify a definition of multi-morbidity as “two or more chronic conditions” in their inclusion criteria but had a sample of participants that were reflective of multi-morbidity, which we felt was the case in these two papers. We recognise this to be a limitation, and as previously discussed, have amended our strengths and limitations section to reflect this.

- Mantelli et al has a clear focus on clinician’s attitude to deprescribing but what is this is not a patient priority?

Response: On full text review on this paper by Mantelli et al, we found the paper reported specific treatment priorities of clinicians, which although were in the context of de-prescribing, we felt still had value in answering our review questions, particularly in understanding the treatment priorities of clinicians. As discussed previously, the difficulty with defining prioritisation required us to make a judgement on the value of each paper in answering our review question, as part of the study selection process, and in this case, we felt that the results of this study did have some value in addressing our aim of understanding the treatment priorities of clinicians.

- Caughey et al examine preferences in the context of scenarios which again is different to actual priorities presenting in clinical practice

Response: We agree Caughey et al have investigated the priorities of patients and clinicians in the context of hypothetical scenarios rather than in “real-life” situations. In our inclusion criteria, we did not make a distinction on whether studies investigated priorities in hypothetical or real-life settings. Also, whilst the priorities were hypothetical, we felt that the results of this study still provided valuable insight into understanding the processes of prioritisation of multi-morbid patients and clinicians.

- Schoenberg – description of the outcome does not seem congruent with the study aims

Response: The description of the outcome for this study given in our table has been updated to better describe how the aim of exploring patient prioritisation was addressed in this study.

- Naik et al- I think this is a very focused population with named specific conditions and not MM-

Response: Thank you, in this case, the authors presented the mean Deyo comorbidity index of their participants, which was 6.85. The Deyo comorbidity index gives a score to the patients’ different chronic health conditions, the sum of which is used to predict “long-term mortality” [17]. As previously discussed, we chose to include studies to also include studies which did not specify a definition of multi-morbidity as “two or more chronic conditions” in their inclusion criteria but had a sample of participants that were reflective of multi-morbidity. While the primary focus in this study was on

patients with cancer, we felt that as the context in this study was prioritisation when faced with a recent diagnosis of cancer but on a background of multi-morbidity, it did address our review question and aim.

- Elliot et al: outcomes seem to be focused on medicines –

Response: We agree the focus from the outcomes seems to be on medicines, however, we felt the aim of exploring “choices about medicines” related to the “treatment priorities” element of our review question, which is why this paper was included in this review.

- Cheraghi Sohi seems to be another qualitative synthesis and would it be more appropriate to screen included studies and see which were eligible rather than having two layers of interpretation-

Response: Thank you, we did consider this and found that in their secondary analysis, Cheraghi Sohi et al extracted and analysed “raw” data from the interview transcripts of selected multi-morbid participants from the list of participants in the original studies, which had not necessarily been presented in the published papers relating to the original studies. Hence, we chose to include the paper Cheraghi-Sohi et al in this study, rather than the original studies.

- Table 1 – might be easier to group similar study types together and address the different types of prioritisation more clearly this way

Response: Thank you, we have re-organised our table according your suggestion.

- Results on disease-specific priorities do not seem as relevant to a review focusing on multimorbidity

Response: We felt that the studies which reported prioritisation of particular conditions in the context of multi-morbidity did still provide valuable insight in addressing the aims of our review, particularly with understanding the how the priorities of multi-morbid patients compare to those of clinicians, as most of these studies reported making a comparison between these two groups. However, we recognise that the term “disease-specific” is misleading, and have therefore changed this to “condition-focused” priorities in our paper.

- The discussion is relatively brief and I think could address some of the issues I have raised above about definitions and process for prioritization

Response: Thank you, as previously discussed, we have updated our discussion section to elaborate on the difficulty with defining prioritisation and how this affected our study selection process.

- The recommendation to include prioritization in future guidelines reflects the author’s lack of awareness of these guidelines. For example, NICE 2016, Ariadne principles, Wallace BMJ and Christiane Muth’s paper on overview of guidelines-

Response: Thank you, Muth et al’s paper regarding the Ariadne guidelines is referenced in our introduction. We are aware of the current NICE guidelines relating to eliciting the priorities, values and goals of multi-morbid patients. We intended to emphasise the need to consider short term priorities well and long term priorities, due to the finding from our review that the processes of prioritisation may differ depending on whether patients were considering the current impact of their illnesses, or the future risk posed by their illnesses. We felt that this would be an additional element to the advice already to clinicians in the current NICE guidelines. However, we acknowledge that this was not clear in our original paper, and we have made amendments to the paper to clarify this point.

“We recommend that future guidelines developed for clinicians in the management of multi-morbidity

highlight the need to elicit and consider both short term and long term priorities for their patients', as our review has shown that patients' priorities for their current illnesses experiences and future risks posed by illnesses, may differ. In accordance with current NICE guidance, we also reiterate the need to review these priorities continually, and particularly when exacerbations, changes to illness course or treatment regimens, or other wider socially-contextualised changes occur in their patients' lives."

References

- 1 van den Akker M, Buntinx F, Knottnerus JA. Comorbidity or multimorbidity: what's in a name? A review of literature, *The European Journal of General Practice* 1996;2:65-70.
- 2 Jowsey T, Dennis S, Yen L, et al. Time to manage: patient strategies for coping with an absence of care coordination and continuity, *Social Health Illn* 2016;38:854-73.
- 3 Mead N, Bower P. Patient-centredness: a conceptual framework and review of the empirical literature, *Soc Sci Med* 2000;51:1087-110.
- 4 Kaba R, Sooriakumaran P. The evolution of the doctor-patient relationship, *International Journal of Surgery* 2007;5:57-65.
- 5 Noel P.H., Frueh B.C., Larme A.C., et al. Collaborative care needs and preferences of primary care patients with multimorbidity. *Health Expectations* 2005;8:54-63.
- 6 Lechner S., Herzog W., Boehlen F., et al. Control preferences in treatment decisions among older adults - Results of a large population-based study. *J Psychosom Res* 2016;86:28-33.
- 7 Noel PH, Parchman ML, Williams JWJ, et al. The challenges of multimorbidity from the patient perspective. *Journal of General Internal Medicine* 2007;22:419-24.
- 8 Coventry PA, Fisher L, Kenning C, et al. Capacity, responsibility, and motivation: a critical qualitative evaluation of patient and practitioner views about barriers to self-management in people with multimorbidity, *BMC health services research* 2014;14:536.
- 9 Mc Namara KP, Breken BD, Alzubaidi HT, et al. Health professional perspectives on the management of multimorbidity and polypharmacy for older patients in Australia, *Age & Ageing* 2017;46:291-9.
- 10 Sinnott C., Mc Hugh S., Boyce M.B., et al. What to give the patient who has everything? A qualitative study of prescribing for multimorbidity in primary care. *British Journal of General Practice* 2015;65:e184-91.
- 11 Luijckx HD, Loeffen MJW, Lagro-Janssen AL, et al. GPs' considerations in multimorbidity management: a qualitative study. *British Journal of General Practice* 2012;62:e503-10.
- 12 Smith SM, Wallace E, Salisbury C, et al. A Core Outcome Set for Multimorbidity Research (COSmm), *Ann Fam Med* 2018;16:132-8.
- 13 Junius-Walker U, Wrede J, Voigt I, et al. Impact of a priority-setting consultation on doctor-patient agreement after a geriatric assessment: cluster randomised controlled trial in German general practices. *Quality in primary care* 2012;20.
- 14 Junius-Walker U, Stolberg D, Steinke P, et al. Health and treatment priorities of older patients and their general practitioners: a cross-sectional study. *Quality in primary care* 2011;19.
- 15 Voigt I, Wrede J, Diederichs-Egidi H, et al. Priority setting in general practice: health priorities of older patients differ from treatment priorities of their physicians, *Croat Med J* 2010;51:483-92.
- 16 Zulman D.M., Kerr E.A., Hofer T.P., et al. Patient-provider concordance in the prioritization of health conditions among hypertensive diabetes patients. *Journal of General Internal Medicine* 2010;25:408-14.
- 17 Ladha KS, Zhao K, Quraishi SA, et al. The Deyo-Charlson and Elixhauser-van Walraven Comorbidity Indices as predictors of mortality in critically ill patients, *BMJ Open* 2015;5:e008990,2015-008990.

VERSION 2 – REVIEW

REVIEWER	Tanisha Jowsey
----------	----------------

	Centre for Medical and Health Sciences Education, University of Auckland, New Zealand
REVIEW RETURNED	04-Dec-2019

GENERAL COMMENTS	Thanks for the opportunity to review this paper. I read it with great interest. The paper is very good. A major limitation of the review was that the term co-morbidity was not searched. The authors mention this in their limitations. I think its a shame. But that aside, the paper is excellent. The authors have detailed their methods, analyses, and findings very well. It is a really useful contribution to the multimorbidity and patient care literature. My comments below are for very minor changes. Best wishes for it, Abstract: 1) Our search identified of 24 studies - delete 'of' 2) 'Our findings have shown that there is a mostly low level of agreement between the priorities of multi-morbid patients and clinicians. We found that prioritisation by multi-morbid patients was mainly driven by their illness experiences, whilst clinicians focused on longer term risks.' - this is a finding, not conclusion. I suggest you move it up. Main text: It seems that for patients the main issue they were concerned with was maintaining independence (which in western cultures has a major bearing on quality of life). I think its worth specifying this in the abstract, since it emerged in both the quant and qual studies. I think its also worth mentioning in discussion (pg16 about line 30) or limitations that the studies included for review are all from western countries. We are not seeing a picture of Asia or the South Pacific, or of Indigenous communities where patient priorities may be very different.
---

REVIEWER	Susan Smith RCSI Ireland
REVIEW RETURNED	26-Nov-2019

GENERAL COMMENTS	Thank you for addressing my previous comments. My only remaining issues which could be considered with the editorial team is the use of the term multi-morbid patient and the hyphenation of multimorbidity/ multimorbid patient. We are aware that patients do not recognise or like the term multimorbidity though it is widely used in research publications and will continue to be used. The MeSH term does not use a hyphen (though the reported search strong for this paper does, which could be a limitation). My recommendation would be to use the term patients with multimorbidity throughout rather than using multimorbid as an adjective. We no longer describe patients as diabetic patients etc .
--

VERSION 2 – AUTHOR RESPONSE

Reviewer: 3

Reviewer Name: Susan Smith

Institution and Country: RCSI, Ireland

Please state any competing interests or state 'None declared': None declared

Please leave your comments for the authors below

Thank you for addressing my previous comments. My only remaining issues which could be considered with the editorial team is the use of the term multi-morbid patient and the hyphenation of multimorbidity/ multimorbid patient. We are aware that patients do not recognise or like the term multimorbidity though it is widely used in research publications and will continue to be used. The MeSH term does not use a hyphen (though the reported search string for this paper does, which could be a limitation). My recommendation would be to use the term patients with multimorbidity throughout rather than using multimorbid as an adjective. We no longer describe patients as diabetic patients etc

Response: We thank the reviewer for their feedback. We have changed our use of the phrase "multimorbid patients" to "patients with multimorbidity" throughout our paper and removed any hyphenation of the term "multimorbidity" (except for hyphenation already present in the titles of included studies). We have also amended the title of our paper to "Priorities of patients with multimorbidity and of clinicians regarding treatment and health outcomes: a systematic mixed studies review".

As detailed in our search string in Appendix 1, we included both hyphenated and non-hyphenated versions of "multimorbidit*" in our searches to ensure that all variations of the term are captured and included "multi morbid*", on advice of our local Academic Librarians, for the same reason. We have updated our key terms in the paper to better reflect our search string.

Reviewer: 2

Reviewer Name: Tanisha Jowsey

Institution and Country: Centre for Medical and Health Sciences Education, University of Auckland, New Zealand

Please state any competing interests or state 'None declared': None declared.

Please leave your comments for the authors below

Thanks for the opportunity to review this paper. I read it with great interest. The paper is very good. A major limitation of the review was that the term co-morbidity was not searched. The authors mention this in their limitations. I think its a shame. But that aside, the paper is excellent. The authors have detailed their methods, analyses, and findings very well. It is a really useful contribution to the multimorbidity and patient care literature. My comments below are for very minor changes.

Best wishes for it,

Abstract:

- 1) Our search identified of 24 studies - delete 'of'
- 2) 'Our findings have shown that there is a mostly low level of agreement between the priorities of multi-morbid patients and clinicians. We found that prioritisation by multi-morbid patients was mainly driven by their illness experiences, whilst clinicians focused on longer term risks.' - this is a finding, not conclusion. I suggest you move it up.

Response: We thank the reviewer for their feedback. We have amended our abstract in accordance with their suggestions:

"Results: Our search identified twenty four studies for inclusion, which comprised of twelve quantitative studies, ten qualitative studies and two mixed-methods studies. Twelve studies reported the priorities of both patients and clinicians, ten studies reported the priorities of patients and two studies reported the priorities of clinicians alone. Our findings have shown a mostly low level of agreement between the priorities of patients with multimorbidity and clinicians. We found that prioritisation by patients was mainly driven by their illness experiences, whilst clinicians focused on longer term risks. Preserving functional ability emerged as a key priority for patients from across our

quantitative and qualitative analyses.

Conclusion: Recognising that there may be a disparity in prioritisation and understanding the reasons for why this might occur, can facilitate clinicians in accurately eliciting the priorities that are most important to their patients and delivering patient-centred care.”

Main text:

It seems that for patients the main issue they were concerned with was maintaining independence (which in western cultures has a major bearing on quality of life). I think its worth specifying this in the abstract, since it emerged in both the quant and qual studies. I think its also worth mentioning in discussion (pg16 about line 30) or limitations that the studies included for review are all from western countries. We are not seeing a picture of Asia or the South Pacific, or of Indigenous communities where patient priorities may be very different.

Response: We thank the reviewer for their feedback. We have amended the “Strengths and limitations” section in the main text of our paper to include the limitation in the global generalisability of our findings due to all of the included studies having been conducted in developed and western countries.

“All of the included studies were conducted in developed, western countries, which limits the global generalisability of our findings, as the priorities of patients with multimorbidity and of clinicians in developing and/or eastern countries may differ to the findings of this review.”